# Deformation and Compressive Strength of Steel Fiber Reinforced MgO Concrete

**DOI:** 10.3390/ma12213617

**Published:** 2019-11-04

**Authors:** Feifei Jiang, Zhongyang Mao, Min Deng, Dawang Li

**Affiliations:** 1State Key Laboratory of Materials-Oriented Chemical Engineering, College of Materials Science and Engineering, Nanjing Tech University, Nanjing 211800, China; mzy@njtech.edu.cn (Z.M.); dengmin@njtech.edu.cn (M.D.); 2College of Naval Architecture Civil Engineering, Jiangsu University of Science and Technology, Zhangjiagang Campus, Suzhou 215600, China; zhjie@just.edu.cn

**Keywords:** concrete shrinkage, MgO expansion agent, steel fiber, compressive strength, self-volumetric deformation, interface structure

## Abstract

To reduce the cracking caused by shrinkage and avoid the brittle behavior of concrete, MgO expansion agent and steel fibers were used in this paper. Firstly, the effect of MgO and steel fibers on the compressive strength of concrete was compared. The results showed that the compressive strength of steel fibers reinforced concrete (SC) and steel fiber reinforced MgO concrete (SMC) was significantly improved. Compared with ordinary concrete (OC), SMC’s 28 days compressive strength increased by 19.8%. Secondly, the influence of MgO and steel fibers with different contents on the self-volumetric deformation of concrete was compared through the experiment. The results showed that as a result of the hydration expansion of MgO, MC and SMC both showed obvious expansion, and their 190 days expansion was 335 με and 288 με, respectively. Lastly, through a scanning electron microscope (SEM) test, it was found that the constraint effect of steel fibers changed the expansion mode of MgO from outward expansion to inward extrusion, thus improving the interfacial bond strength of concrete.

## 1. Introduction

Owing to its high compressive strength, good corrosion resistance, low price, and excellent operating performance, concrete has become the most popular building material all over the world. However, crack is a fatal flaw in concrete, which not only reduces the mechanical properties of concrete, but also provides channels for the intrusion of harmful substances in the outside world. Moreover, cracks accelerate the corrosion and destruction of concrete and seriously affect the durability of concrete, especially in ports and Marine structures.

In order to restrain concrete cracking and enhance its durability, researchers have proposed three methods. One is to compensate for shrinkage of cement-based materials during hydration by adding expansion agents, such as CaO-type, MgO-type, and AFt-type expansion agents, which can reduce shrinkage cracks [1,2,3,4,5]. The other is to improve the crack resistance of concrete by adding fibers, such as polypropylene (PP) fibers, polyvinyl alcohol (PVA) fibers, and steel fibers, which can reduce plastic cracks and improve the ductility of concrete [6,7,8,9]. However, there are some shortcomings in these two methods. They can reduce shrinkage cracks by using an expansion agent, but they have no effect on cracks under load. What is more, only after concrete cracks can fibers play a bridge role, and they can only refine cracks, but not prevent them from occurring [10,11]. For fiber-reinforced expansive concrete, with concrete as a carrier, an expansion agent as the expansive element, and three-dimensional fibers distributed randomly as constraints, composite materials are assembled. This method can overcome the shortcomings of expansion agent and fiber used alone and give full play to the two advantages. Besides, it can even produce the “superposition” effect, effectively improving the crack resistance of concrete [12,13,14,15].

In the past two decades, some researches have focused on the engineering application of fiber-reinforced concrete and expansive concrete, such as railway sleepers [16,17,18], dams [19], and airport pavements [20]. In these studies, Ahsan Parvez proposed a method to improve the physical performance of the concrete sleeper such as durability, energy absorption, fatigue, and tolerance with the inclusion of steel fibers, and he found that a minimum volume of fibers (0.25%) was essential to ensure enhanced performance [16]. Kaijian Huang proposed a method to enhance the volume stability of airport pavements in high altitude localities by adding fibers and MgO expansion agent [20].

In this paper, the authors combined MgO expansion agent with steel fibers. The expansion behavior of MgO is restrained by steel fibers in the hardening process and produces self-pressure, which can significantly improve the cracking load. The large vertical pressure exists at the interface between the cement matrix and steel fibers, which significantly improves the bonding property. Compared with polypropylene (PP) fibers and polyvinyl alcohol (PVA) fibers, steel fibers have a higher modulus of elasticity, and thus have stronger binding capacity, which can significantly improve the bonding property.

Owing to the constraints of steel fibers, the expansion and mechanical properties of steel fiber-reinforced MgO concrete (SMC) are quite different from those of MgO concrete (MC). In this paper, self-volumetric deformation experiments and compressive experiments were performed to reveal the expansion property of MC and SMC and the mechanical properties of SMC, which can help us understand SMC better and provide a theoretical basis for further engineering application.

## 2. Materials and Experiments

### 2.1. Materials

Portland cement, fly ash, coarse aggregates, fine aggregates, steel fibers, and polycarboxylate based plasticizer were used as the experimental objects. The details about each material are enumerated below.

#### 2.1.1. Cement

The cement was produced by Shandong Shan Aluminum Cement Co., Ltd. (Zibo, China). The specific model was Shan Lu P.0. 52.5 ordinary Portland cement. The 3 days compressive strength of cement was 33.2 MPa and the 28 days was 59.8 MPa. The initial setting time of cement was 220 min, and the ultimate setting time was approximately 290 min. Table 1 shows the detailed chemical composition of the cement.

#### 2.1.2. Fly Ash

Fly ash was Class I fly ash produced by Shenhua Huashou Power Co., Ltd. (Shanghai, China). The sulfur trioxide content of fly ash was 1.2% and the water storage ratio was 74%. Table 2 shows the detailed chemical composition of the fly ash.

#### 2.1.3. Steel Fiber

Steel fibers were wavy steel fibers produced by Shuanglian Building Materials Co. LTD in Zibo, China and the tensile strength of the steel fibers was 520 MPa. The diameter of the steel fibers was 0.58 mm and the length was 38 mm. The photograph of the wavy steel fibers utilized in the present study is depicted in Figure 1.

#### 2.1.4. MgO Expansion Agent

MgO expansion agent was produced by Wuhan Sanyuan Special Building Materials co. LTD in Wuhan, China, and its activity was 115 s. The crystal size of MEA was 36.3 nm and the specific surface area was 45.7 m^2^/g. The photograph of MgO in the present study is depicted in Figure 2. Table 3 shows the detailed chemical composition of MgO.

#### 2.1.5. Mix Proportions

The mix proportions used in the present study are listed in Table 4. The content of MgO expansion agent was 8% of the total amount of cementitious materials, by weight. Besides, the content of steel fibers was 1%, by volume.

### 2.2. Experimental Works

#### 2.2.1. Strain Gauge

Type-VWS-10 vibrating wire strain gauges made by Nanjing Gelan Industrial Co., Ltd. (Najing, China) were used. The gauge distance of strain gauge was 100 mm, and the vibrating modulus of the strain gauge, F, was closely related to its length, varying with the length change of the transducer brought about by deformation of concrete [20]. The strain was calculated by Equation (1):
(1)ε=k∆F+(b−a)∆T
where ε is the self-volumetric deformation of concrete; *k* is measurement sensitivity of strain gauge; ∆F is the change in the measured value in real time relative to the base value of vibrating modulus; *b* is temperature correction factor of strain gauge; a is the linear expansion coefficient of concrete; and ∆T is the change of the temperature.

#### 2.2.2. Experiment Works

For each mix design, nine cylinders (dimension ∅50 mm × H100 mm) were cast for the measurements of compressive strength (ASTM C109 [21]), at 3 and 28 days after casting. In addition, four plastic buckets were used to test the self-volumetric deformation of concrete. The size of each plastic bucket was ∅250 mm × 300 mm (Figure 3). The concrete was poured into a plastic bucket and the strain gauge was placed vertically in the bucket at the same time. Then, the concrete surface was sealed with epoxy resin to prevent moisture exchange. After that, the plastic bucket was moved to a 20 degree curing room to maintain a constant temperature and absolute humidity (Figure 4). All test preparations were to be completed within 20 min. Lastly, the strain of concrete was tested every hour to research the self-volumetric deformation of concrete.

#### 2.2.3. Microstructure Characterization

In order to study the effect of the steel fibers and MgO expansion agent on the microstructure of concrete, fractured surfaces of concrete at the age of 30 days were investigated by scanning electron microscope (SEM, JSM-6510LA, JEOL, Ltd., Tokyo, Japan).

## 3. Results and Discussion

### 3.1. Failure Pattern of Concrete

The process of concrete compression failure is actually the process of crack generation, crack propagation, and crack penetration. In the process of loading, each specimen had experienced the elastic stage, elastic-plastic stage, and failure stage. Figure 5 shows the different failure modes of four different concrete specimens at 28 days. The failure modes of OC and MC specimens were brittle failure. Before the failure, a main crack first appeared on the surface of the specimen, with a wide width and few associated cracks. Moreover, the main crack developed rapidly, and the angle between the inclined crack surface and the vertical line of the load was 60–70 degrees.

On the other hand, the failure modes of SC and SMC specimens were obviously brittle. When the specimen approached the peak stress gradually, a small number of small vertical cracks began to appear on the surface of the specimen. After reaching the peak point, the bearing capacity of the specimen decreased more and more slowly, and a large number of discontinuous longitudinal short cracks appeared. Finally, the specimen could not return to the original deformation state and formed an inclined main crack. The angle between the inclined crack surface and the vertical line of the load was 70–80 degrees, and the failure form of steel fibers was pull out failure.

### 3.2. Compressive Strength of Concrete

Figure 6 presents the results of the 3 days and 28 days compressive strength of the concrete specimens. For OC, the 3 days and 7 days compressive strength was 47.3 MPa and 58.7 MPa, respectively. For MC, the 3 days and 7 days compressive strength of concrete increased by 1.69% and 5.79% respectively, to 48.1 MPa and 62.1 MPa. The reason for this increase is the addition of MgO, which increases the total amount of active ingredients. For SC, the 3 days compressive strength of concrete decreased by 2.11%, because the addition of steel fibers leads to an increase in the number of interfaces between fiber and cement matrix, and the 28 days compressive strength of concrete increased by 9.54%, because the steel fibers have the effect of preventing crack development.

For SMC, the 3 days and 7 days compressive strength of concrete increased by 4.23% and 19.8% respectively, to 49.3 MPa and 70.3 MPa. There are two main reasons for the rise in the strength of SMC. First, steel fibers prevent cracks from developing, and delay further damage to concrete. Secondly, the combined use of MgO and steel fibers produces a “superstack” effect and, under the constraint of steel fibers, and the stress of concrete becomes a three-way compression state, which create self-pressure inside the concrete and further increases the strength of the concrete.

### 3.3. Deformations of Concrete

The strain of concrete with different dosages of steel fibers and MgO, which represents the self-volumetric deformation, is listed in Figure 7. For OC, its early shrinkage rate was relatively large (0–40 days), reaching 165 με, after which the shrinkage rate was significantly reduced, and its 190 days shrinkage strain was 178 με. For SC, its contraction curve was very similar to OC; however, different from OC, the late shrinkage of SC decreased significantly, and its 190 days shrinkage decreased by 24.1% to 135 με. This means that steel fibers can improve the volumetric stability of concrete to some extent owing to the restraint effect of steel fibers. For MC, the specimen showed an obvious expansion owing to the addition of MgO, and the expansion value of 190 days reached 335 με, which means that MgO can effectively compensate for the shrinkage of concrete. Because of the large shrinkage of concrete in the early stage (0–40 days), the expansion of MC was small in the early stage and gradually increased in the late stage.

The expansion curve of SMC was similar to that of MC, but the 190 days expansion value of SMC was 16.3% less than that of MC owing to the constraint action of steel fibers, reaching 288 με. In the first stage (0–50 days), the SMC expansion energy was almost all used for tensioning steel fibers because of the strong constraint effect of steel fibers. Therefore, the total volume expansion of SMC was very small. The 50 days expansion was only 25 με, which was 50% of the expansion of MC. After that, the expansion of SMC continued to increase as the restraint of the steel fibers weakened significantly, and both steel fibers and concrete matrix expanded outward together.

### 3.4. Performance Enhancement Mechanism of Concrete

#### 3.4.1. The Relationship between Expansion and Stress

Figure 8 shows different expansion models of the concrete specimens. For expansion in the free state, the adjacent particles of MC move away from each other in a back direction, thus generating tensile stress in the concrete. What is more, in the process of concrete hardening, the compactness of concrete may be reduced owing to the existence of tensile stress. When the tensile stress exceeds the ultimate tensile stress of concrete, the concrete will crack. Therefore, when the expansion agent is used in excess, greater harmful expansion will also cause the concrete to crack [4]. On the other hand, for expansion under the constraint condition of steel fibers, the adjacent particles of SMC move in the opposite direction and get close to each other, so as to generate compressive stress in the concrete, that is, chemical pre-pressure, which indirectly improves the tensile strength of concrete.

#### 3.4.2. The Interface Structure of Concrete

Concrete interface is the weak link of the concrete specimens, and the interface between cement matrix and aggregate directly affects the mechanical properties and durability of concrete [22,23,24]. Scanning electron microscope was performed to study the effect of steel fibers and MgO expansion agent on the interface. Figure 9 reflects the interface morphology of cement matrix and aggregate in concrete. For MC specimen in Figure 9a, there were many defects in the interfacial transition zone, long cracks in the parallel direction, and many associated cracks in the vertical direction. On the other hand, for SMC specimens in Figure 9b, because MgO and steel fibers act as extrusion fillers and pore refiners, a large vertical pressure exists at the interface between cement matrix and steel fibers, and the transition zone of SMC becomes very dense. In the vertical direction of SMC, only a few small cracks can be found, and the interface is intact, indicating that they have better interfacial bonding strength.

## 4. Conclusions

Considering the defects of MgO expansion agent or steel fibers when used alone, this paper combined MgO expansion agent and steel fibers. On the basis of detailed experiments and theories, the following conclusions can be drawn:

(1) After the combined use of MgO expansion agent and steel fibers, the damage pattern of SMC under compression changed significantly compared with OC, and the SMC showed obvious ductile damage. The compressive strength of SMC increased by 19.8%, which was much greater than the simple sum of strength increments of MC (5.79%) and SC (9.54%).

(2) Compared with OC, the 190 days shrinkage of SC was reduced by 24.1% owing to the restraint effect of steel fibers. However, the steel fibers cannot completely offset the shrinkage of the concrete, and the 190 days shrinkage of SC still reached 135 με. Owing to the hydration of the MgO expansion agent, MC and SMC showed significant expansion, and their 190 day expansion was 335 με and 288 με respectively. Further, the steel fibers also restrained the expansion, causing the SMC’s expansion to be 16.3% less than MC’s.

(3) According to the SEM test, there were many defects in the interface between cement matrix and aggregate in MC, while SMC had a denser interface and better bond strength, which explained why the compressive strength of SMC was significantly higher than MC.

(4) Thanks to the positive results in this study, SMC should be widely used in the basement, bridge deck, and other structures with high requirements for anti-seepage and crack resistance.

## Figures and Tables

**Figure 1 materials-12-03617-f001:**
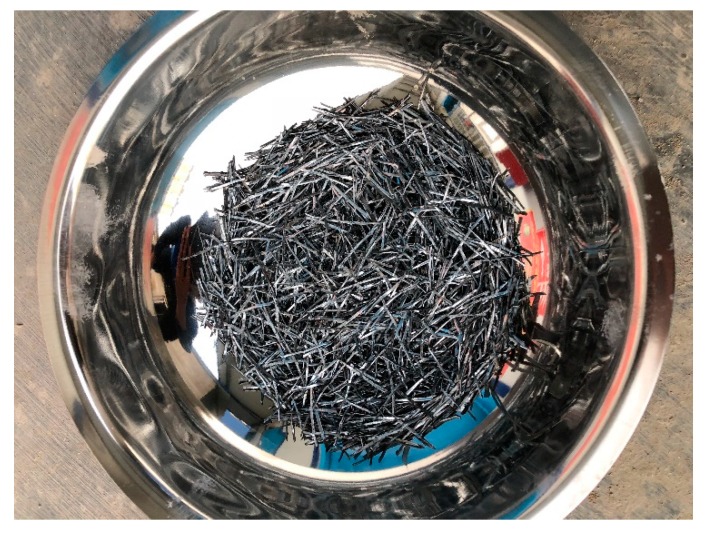
Photograph of wavy steel fibers.

**Figure 2 materials-12-03617-f002:**
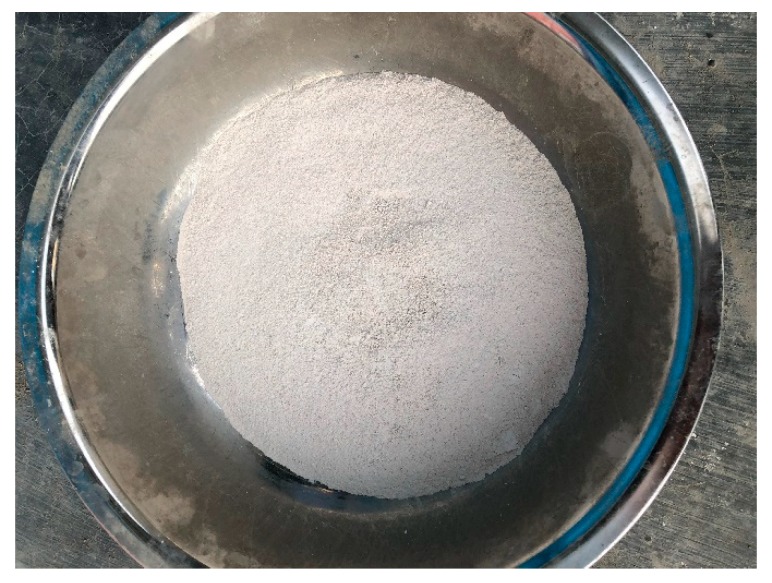
Photograph of MgO.

**Figure 3 materials-12-03617-f003:**
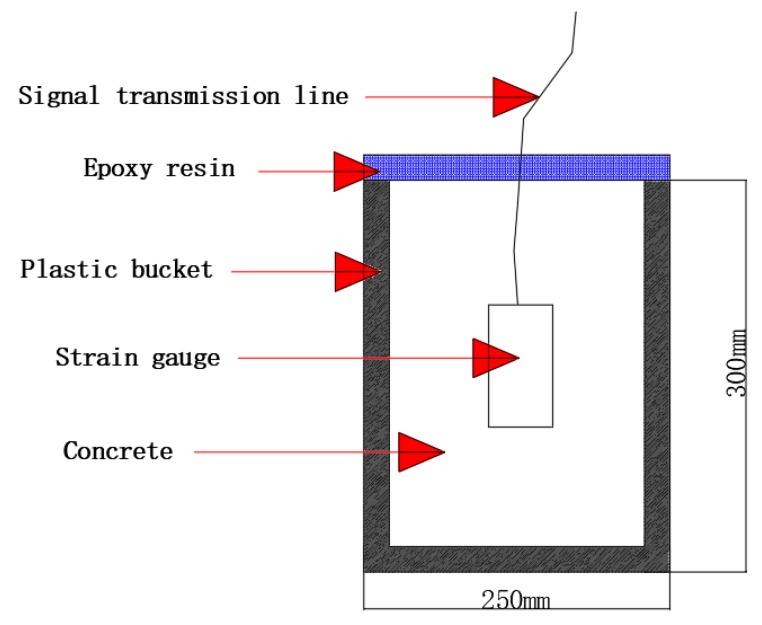
Schematic diagram of measuring device using strain gauge.

**Figure 4 materials-12-03617-f004:**
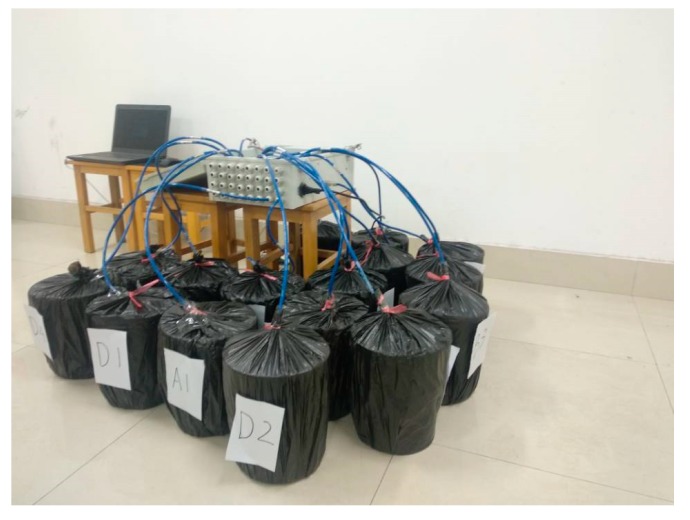
Self-volumetric deformation testing device for concrete.

**Figure 5 materials-12-03617-f005:**
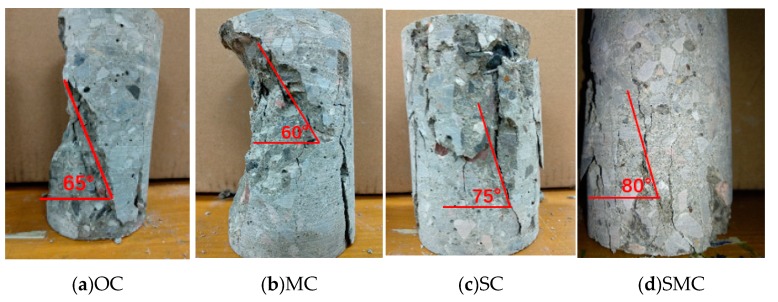
Failure pattern of concrete. (**a**) OC, ordinary concrete; (**b**) MC, MgO concrete; (**c**) SC, steel fiber-reinforced concrete; (**d**) SMC, steel fiber-reinforced MgO concrete.

**Figure 6 materials-12-03617-f006:**
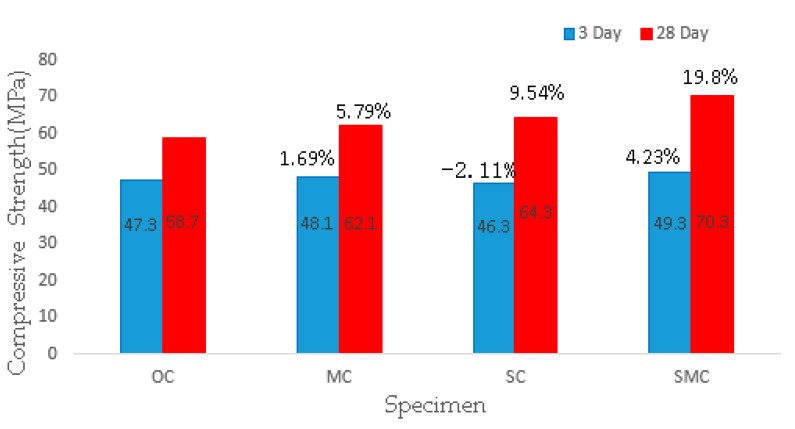
Compressive strength at 3 and 28 days.

**Figure 7 materials-12-03617-f007:**
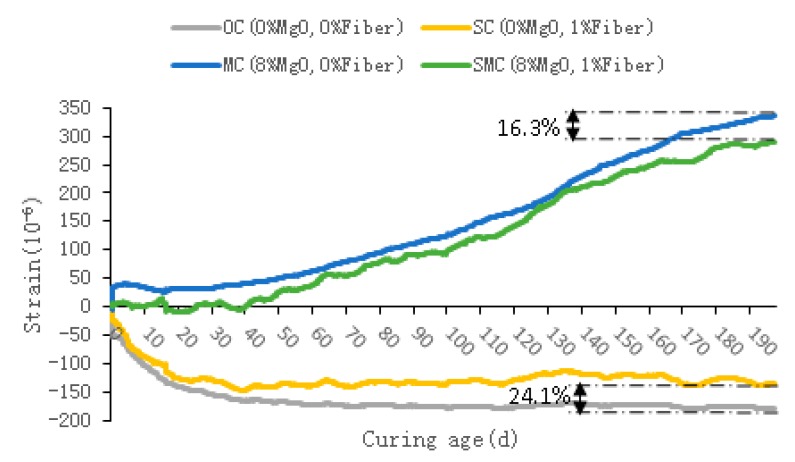
Self-volumetric deformation of concrete.

**Figure 8 materials-12-03617-f008:**
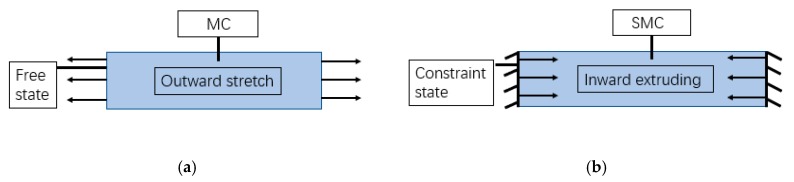
Expansion model of concrete: (**a**) Expansion under freedom, (**b**) Expansion under constraint.

**Figure 9 materials-12-03617-f009:**
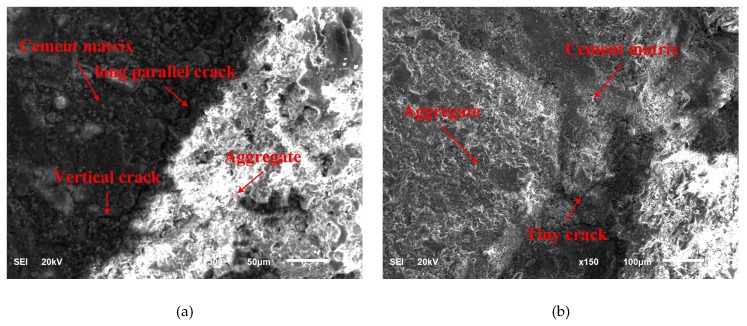
Scanning electron microscope (SEM) image of interface between cement matrix and aggregate: (**a**) MC, (**b**) SMC.

**Table 1 materials-12-03617-t001:** Chemical composition of cement.

Type	Chemical Composition/wt %									
	CaO	MgO	Al_2_O_3_	SiO_2_	Fe_2_O_3_	SO_3_	K_2_O	Na_2_O	Loss	Total
Cement	60.51	2.18	6.34	22.02	3.05	1.86	0.47	0.23	1.96	98.62

**Table 2 materials-12-03617-t002:** Chemical composition of fly ash.

Type	Chemical Composition/wt %									
	CaO	MgO	Al_2_O_3_	SiO_2_	Fe_2_O_3_	SO_3_	K_2_O	Na_2_O	Loss	Total
Fly ash	5.01	1.03	34.18	48.91	5.22	1.20	0.89	0.62	1.50	98.56

**Table 3 materials-12-03617-t003:** Chemical composition of MgO.

Type	Chemical Composition/wt %						
	CaO	MgO	Al_2_O_3_	SiO_2_	Fe_2_O_3_	Loss	Total
MgO	3.19	85.44	0.73	4.45	0.42	4.49	98.72

**Table 4 materials-12-03617-t004:** Mix proportion of concrete.

Specimen	Composition/kg·m^−3^								W/C
	Cement	Fly Ash	Fine Aggregate	Coarse Aggregate	Water	Water Reducer	Steel Fiber	MgO	
ordinary concrete (OC)	450	50	713	1025	160	6	0	0	0.32
MgO concrete (MC)	450	50	713	1025	160	7.5	0	40	0.32
steel fiber-reinforced concrete (SC)	450	50	713	1025	160	6	78	0	0.32
steel fiber-reinforced MgO concrete (SMC)	450	50	713	1025	160	7.6	78	40	0.32

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
