# Peer review of "Deformation and Compressive Strength of Steel Fiber Reinforced MgO Concrete"

_materials, 2019, doi:10.3390/ma12213617_

Round 1

Reviewer 1 Report

Comments

This paper investigated the hybrid concrete. This is an interesting study and the reviewer can recommend for publication if the author satisfactorily address the following comments in the revised version.

The author need to present compressive load vs displacement or stress vs strain behaviour to confirm the ductile behaviour for SMC. Why SMC had a denser interface and better bond strength need to briefly explain in third conclusion.

What was the particle size for MgO? This is important to mention as the workability depends on it.

Table 1, Table 2 and Table 3 can be combined into one Table for better understanding. “The meter of steel fibers is 0.58mm….” It should be diameter. The word “superstack effect” need to be defined.

Introduction section need to be improved further. Hybrid concrete has lot of applications including the manufacturing of railway sleepers. The application need to be highlighted in the introduction section. The following paper can be referred “Hybrid FRP-concrete railway sleeper” and “A detailed procedure of mix design for fly ash based geopolymer concrete” where it is showing the application.

Reviewer 2 Report

The authors present a work on the Deformation and compressive strength of steel fiber reinforced MgO concrete. The subject of the authors work is an important significant issue in structural engineering and materials, and such an attempt is of great interest. 

At the beginning of the article, the authors reviewed the literature on the subject of research. It is worth noting that many literature items are very new. However, in my opinion, this review could be broader and contain more details from the studies of other authors. It would be good to show a summary of the results of other researchers (e.g. in a table or graphs). It would later be a reference point for the research contained in the reviewed article.

The scope and description of the research presented below is comprehensible and factually correct. The research methods used are simple and widely known. Therefore, they do not require special description. The analysis of the results obtained is also described correctly.

I think that the results obtained by the authors are important from the point of view of construction practice. However, the summary does not link these results to construction practice. It would be worth pointing out where the concrete designated as SMC could be used. Experimental research should make a significant contribution to learning and to practice.

However the paper, in its present form, requires some substantial modifications in order to justify its publication in an International Journal such as Materials. I think that the following points should be further elaborated by the authors:

please extend the study part (provide more detailed information from the results of other researchers cited in this article), photographs shown in Figure 9 are blurred. Maybe you can give better quality, please indicate the practical application of concrete designated as SCM in construction.

I have no substantive comments.
